# Laser-Assisted Fabrication for Metal Halide Perovskite-2D Nanoconjugates: Control on the Nanocrystal Density and Morphology

**DOI:** 10.3390/nano10040747

**Published:** 2020-04-14

**Authors:** Athanasia Kostopoulou, Konstantinos Brintakis, Efthymis Serpetzoglou, Emmanuel Stratakis

**Affiliations:** 1Institute of Electronic Structure and Laser, Foundation for Research and Technology—Hellas, 71110 Heraklion, Crete, Greece; kbrin@iesl.forth.gr (K.B.); eserpe@iesl.forth.gr (E.S.); 2Department of Physics, University of Crete, 71003 Heraklion, Crete, Greece

**Keywords:** 2D materials, laser-induced synthesis, anion exchange, nanoparticles, graphene oxide, photo-induced processes, synergistic effects

## Abstract

We report on a facile and rapid photo-induced process to conjugate graphene-based materials with metal-halide perovskite nanocrystals. We show that a small number of laser pulses is sufficient to decorate the 2-dimensional (2D) flakes with metal-halide nanocrystals without affecting their primary morphology. At the same time, the density of anchored nanocrystals could be finely tuned by the number of irradiation pulses. This facile and rapid room temperature method provides unique opportunities for the design and development of perovskite-2D nanoconjugates, exhibiting synergetic functionality by combining nanocrystals of different morphologies and chemical phases with various 2D materials.

## 1. Introduction

Nanoconjugates consisting of nanocrystals coupled with graphene-based materials have recently attracted the interest of the scientific community for their synergistic properties originating from the coupling of the two different materials [1,2,3,4,5,6,7]. The functional groups (hydroxy, carbonyl, carboxy) on their basal planes or edges of the 2D materials—which make them soluble in multiple solvents, coupled with their low dimensionality—offer attractive properties for applications ranging from photovoltaics to photodetectors and sensors [8,9,10,11]. The coupling of these 2D materials with semiconducting quantum dots, mainly synthesized in colloidal solutions, explores the usage of these materials in light-harvesting [12], as catalysts for oxygen reduction in fuel cells [2], electrocatalysts for water splitting [5], as well as in Li-ion batteries [3] derived from the energy and charge transfer, [4,13] or synergetic and enhanced properties [5,14] of the complex structures.

In recent years, the plethora of synthesis pathways for the fabrication of lead halide perovskite nanocrystals has given rise to their conjugation with 2D counterparts, in order to explore new synergetic phenomena [14,15,16,17,18,19]. The metal halide perovskite nanocrystals are a new class of nanomaterials; their properties can be modified by tuning the halide component [20], or by altering their dimensionality or morphology [21,22]. Anion exchange reactions can be used to change the optical properties of these nanostructures. The ions Cl^−^ and I^−^ can partially or totally substitute Br^−^ in the crystal structure of the CsPbBr_3_ through a chemical anion exchange reaction in solution [20,23] or by a photo-induced process in a Cl^−^ or I^−^ containing solvent [24]. Furthermore, in such thin nanocrystals, the optical properties may be tuned by modifying the thickness of them (nanoplatelets) [25,26].

Different morphologies of metal halide perovskite nanocrystals, such as nanospheres [15], nanocubes [1,16,17,18,27] or nanowires [14] have been grown on 2D materials and utilized for applications in light harvesting [14], for photocatalytic CO_2_ reduction [15,17], for visible-light photocatalyst for H_2_ evolution in aqueous HI solution [28] and in photoelectric detection [16]. The multiple applications of the perovskite-2D nanoconjugates lead the way to find easy, fast, solution-processable and controllable methods to fabricate such complex structures with attractive properties. It is important to note here that the above nanoconjugates are synthesized by hot-injection or re-precipitation chemical methods at high [15,16] or mild [17] temperatures in which the 2D materials are presented during the perovskite nanocrystal synthesis. These syntheses need complex apparatus such as Schleck line and give uncontrollable density and non-uniform nanocrystals on the 2D-materials. As a result, the properties of the perovskite-2D nanoconjugates could be varied from one batch to another.

This work uses a completely different approach to anchor previously fabricated metal-halide perovskite nanocrystals on graphene-based materials. A laser-assisted method was used for the first time for the fabrication of perovskite-2D nanoconjugates in colloids. It is shown that a few femtosecond pulses suffice to decorate the flakes with metal-halide perovskite nanocrystals without the initial nanocrystal morphology to be affected. The nanocrystals density can be finely controlled by the number of irradiation pulses. First, the nanocrystals are decorated at the periphery of the flakes (number of pulses <1000) and then are attached on the basal plane of them (number of pulses >1000). This facile and rapid room temperature method provides unique opportunities for the cost-effective and large-scale synthesis of fully controllable perovskite-2D nanoconjugates by combining nanocrystals of different morphologies and chemical phases together with multiple 2D materials.

## 2. Materials and Methods

### 2.1. Preparation of the Perovskite–GO Nanoconjugate Colloid

A published protocol from our group based on a precipitation colloidal method was used for the synthesis of the metal halide nanocrystals [29]. 0.4 mmol PbBr_2_ (trace metals basis, 99.999%, Sigma-Aldrich Corporation, St. Louis, MO, USA) and 0.4 mmol CsBr (anhydrous, 99.999%, Sigma-Aldrich Corporation, St. Louis, MO, USA) were dissolved in 10 mL of DMF (anhydrous, 99.8%, Alfa–Aesar, Haverhill, MA, USA) and left under stirring for three hours in the protective atmosphere of a glovebox. Then, 1 mL of oleic acid (technical grade, 90%, Alfa Aesar, Haverhill, MA, USA) and 0.5 mL oleylamine (approximate C18-content 80 – 90%, ACROS Organics, Geel, Belgium) were added in the above solution. All the chemicals were used without further treatment, except oleic acid which was degassed for 1 h at 120 °C. Afterwards, 0.9 mL of the precursor solution was added rapidly in 10 mL of anhydrous toluene (10 mL) in a sealed vial which was placed in ice. The solution was left under vigorous stirring (1000 RPM) for 30 minutes. Subsequently, it was retained on the bench for a week. Then, perovskite nanocrystals were separated upon centrifugation at 1000 RPM for 5 min and finally re-dispersed in 1,2–dichlorobenzene (DCB, spectrophotometric grade, 99%, SigmaAldrich Corporation, St. Louis, MO, USA). Then, this solution was mixed with the DCB-based solution of the graphene oxide to proceed with the photo-induced process. GO was prepared from graphite powder according to a modified Hummers’ method [30].

### 2.2. Photo-Induced Process for the Fabrication of the Perovskite–GO Nanoconjugates

The laser setup for the fabrication of the nanoconjugates comprised an Yb:KGW ultrafast pulsed laser source, two mirrors and a convex lens of 20 cm focal length (Figure 1). The experiments were conducted at ambient conditions at room temperature. The laser source produced linearly polarized pulses with 170 femtosecond pulse duration and was adjusted to generate 513 nm laser wavelength at 60 kHz repetition rate. The quartz cuvette with the solution was fixed at 5 cm out-of-focus distance for all the irradiations. The Gaussian spot diameter was 700 μm at 1/e^2^, which was measured and analyzed by using a CCD camera. The laser fluence was stable for all the experiments at the value of 0.5 mJ/cm^2^ while the number of the pulses was changing in a wide spectrum, starting from a single until 115.2 million pulses.

### 2.3. Characterization of the Materials

Low magnification and HRTEM images were recorded on a LaB6 JEOL 2100 transmission electron microscope (JEOL Ltd, Akishima, Tokyo, Japan) operating at an accelerating voltage of 200 kV. All the images were recorded by the Gatan ORIUS TM SC 1000 CCD camera (Gatan Inc. Pleasanton, CA, USA). For the TEM analysis, a drop of the as-prepared and the irradiated DCB-based solution was deposited onto a carbon-coated copper TEM grid, which was eventually evaporated. The structural features of the nanocrystals were studied through FFT patterns obtained from the HRTEM images. About 130 individual nanohexagons were counted up for size distribution diagrams and for calculation of the nanocrystal mean size (diagonal length).

XRD studies were performed on a Rigaku D/MAX-2000H rotating anode diffractometer (Rigaku, Tokyo, Japan) with Cu Kα radiation, equipped with a secondary graphite monochromator. The XRD data were collected at room temperature over a 2θ scattering range from 10° to 40°, with a step of 0.02° and a counting time of 20 s per step. The as-prepared and the irradiated samples were dried before the XRD experiments.

The UV-Vis absorption spectra of the solutions before and after laser irradiation were collected at room temperature on a Perkin Elmer, LAMBDA 950 UV/VIS/NIR spectrophotometer (PerkinElmer Inc., Waltham, MA, USA). The solutions were placed in quartz cuvettes.

The fluorescence emission of the solutions placed in quartz cuvette were measured at 300 K on a Fluoromax-P Phosphorimeter (Horiba Ltd., Kyoto, Japan) employing a 150 W Xenon continuous output ozone-free lamp.

Raman spectroscopy was performed on the GO solutions after evaporation of a small quantity using a Nicolet Almega XR Raman spectrometer (Thermo Fisher Scientific, Waltham, MA, USA) with a 473 nm blue laser as an excitation source.

## 3. Results and Discussion

A facile low temperature precipitation-based method was utilized for the synthesis of the metal halide perovskite nanocrystals. This synthesis protocol was published earlier from our group and these nanomaterials tested for their effectiveness as anode materials in Li–air batteries utilizing aqueous electrolyte [29]. Metal halide nanohexagons of diagonal length L = 93.5 ± 11.5 nm were successfully synthesized, centrifuged and re-dispersed in dichlorobenzene (DCB). HRTEM images, together with the analysis of their respective FFT pattern of an individual nanohexagon, revealed the high-crystallinity of these nanocrystals (Figure 2a). The crystal structure of the nanohexagons was compatible with the rhombohedral Cs_4_PbBr_6_ crystal structure (ICSD-025124) (Figure 2a upper inset). Furthermore, weaker diffraction spots were observed in the FFT pattern which coincide with the (200) crystal planes (corresponding to lattice fringes with interplanar spacings of 0.4 Å) and with the (110) (interplanar spacings 5.4 Å) of the orthorhombic CsPbBr_3_ crystal structure (ICSD-97851). The coexistence of two phases of different crystal structure was also confirmed by the analysis of the XRD patterns (Figure 2b).

The solution of the nanohexagons was PL-active under UV illumination (Figure 2a, bottom inset). The origin of the bright green emission in Cs_4_PbBr_6_ nanocrystals is still under debate [31]. Two explanations have been given in the literature for this origin. In the first, the green emission was originated from the coexistence of two phases upon synthesis (CsPbBr_3_ and Cs_4_PbBr_6_) [32]. In the second explanation, this emission was attributed to point-defects emission originating from the intrinsic features or bromide vacancies [33] in the non-fluorescent 0D perovskite structure [34]. Our indication is the first experimental observation that the strong photoluminescence of the Cs_4_PbBr_6_ nanohexagons arises from the inclusion of a fluorescent phase in a non-fluorescent similar to that observed in green emitting Cs_4_PbBr_6_ powders [35] or due to synergetic effects at CsPbBr_3_–Cs_4_PbBr_6_ interfaces [36].

Graphene oxide (GO) was also exfoliated in the same solvent (Figure 3a). The dichlorobenzene solvent was adopted to enable effective exfoliation of the GO with no effect on the perovskite nanocrystals. The nanocrystals were well-dispersed in the same solvent (Figure 3b). *N*,*N*–dimethylformamide (DMF) was tested also as dispersive solvent due to its appropriateness for exfoliation, but this had a strong influence on the perovskite nanocrystals (Appendix A). As a consequence, the DMF was not a good solvent for the fabrication of the nanoconjugates. The DCB-based solutions of the GO and of the metal halide perovskite nanocrystals were mixed and placed in a quartz cuvette.

Some of the nanocrystals were attached on the periphery of the GO flakes (Figure 3c) and most of them remain individually in the solution when the two solutions were mixed (Appendix A). The partial attachment of the nanocrystals on the GO flakes was also confirmed by the photoluminescence (PL) spectra (Figure 3d). In the case of the individual nanocrystals, a single peak was observed and centered at 516 nm (Figure 3d, green curve). The narrow size distribution of the nanocrystals was indicated from the narrow PL full-width half maximum (FWHM). Furthermore, when the nanocrystals were anchored on GO, the peak was blue shifted for only 4 nm, but the intensity was decreased significantly (Figure 3d, red curve), which was also observed in similar nanoconjugates synthesized chemically [15,16,28]. Both solutions have strong and narrow absorption peaks at 311–315 nm and an absorption onset at 512 nm (Appendix A). This absorption features were in agreement with previous reports on nanocrystals of Cs_4_PbBr_6_ with the presence of a secondary CsPbBr_3_ chemical phase [37,38]. Furthermore, the sharp dip at around 330 nm has been observed in systems where these phases coexist [38,39]. These main features remain when the perovskite solution was mixed with the GO solution.

Then, a photo-triggered process was utilized to see if nanoconjugates with higher density of nanocrystals decorated on GO flakes could be obtained. A high repetition rate femtosecond laser system using a directly diode-pumped Yb:KGW (ytterbium doped potassium gadolinium tungstate) as active medium, was employed for the irradiation of the mixed perovskite–GO solution (Figure 1). The laser wavelength, the repetition rate and the pulse duration used for that purpose were 513 nm, 60 kHz and 170 fsec, respectively. Τhe laser fluence in all the experiments kept constant at the value of 0.5 mJ/cm^2^ and only the number of the pulses was varied. The colloids obtained by this process, were stable and no precipitation was observed. The influence of the pulses number, ranging from 1 to 10^6^, on the morphologic characteristics of the nanoconjugates were examined by TEM (Figure 4a–g). The density of the nanocrystals was increased with the number of the pulses retaining their primary size (Appendix A) and the perovskite nanocrystals were assembled first on the periphery of the GO flakes and then on the basal plane when the number of the pulses was increased to over 10^3^ pulses. The peak of the PL of the irradiated solutions was located at the same wavelength with that of the simply mixing, but the intensity was reduced as the number of the pulses increases (Figure 4h). The same wavelength of the PL peak indicated that no anion exchange takes place for these irradiation conditions.

The scheme in the Figure 5 represents the proposed mechanism. The GO includes functional groups (hydroxyl, carbonyl, carboxylic) on their basal planes or edges which could be functional for nanocrystal anchor. Raman spectra indicated that no complete reduction of the as-prepared GO occurs in these irradiation conditions. The D/G intensity ratio (I_D_/I_G_) was 0.72 and 0.70 before and after laser-irradiation with 3.6 million pulses (Appendix A). This ratio decreases to 0.66 for even larger number of pulses (28.8 million pulses). According to previous reports, the carboxyl groups on the edges were coordinated mainly with Pb^+2^ ions of the metal halide nanocrystals crystal structure [40,41].

The effect of the number of the irradiation pulses to the final morphology of the nanoconjugates was drawn from the careful observation of the TEM images (Figure 4 for 1 to 10^6^ irradiation pulses and Appendix A for a larger number until 57.6 × 10^6^ pulses). The perovskite nanocrystals remain on the periphery of the GO flakes for the irradiation duration from 1 to 100 pulses (Figure 4a–c) while for 10^3^ to 10^4^ pulses the density of the nanocrystals on the GO flakes increases significantly (Figure 4d–f). For irradiation of the 10^6^ pulses, the nanocrystals were starting to detach from the basal planes of the flakes (Figure 4g, Appendix A) and remain mainly around the periphery (Appendix A) while in the case of 1.8 × 10^6^ pulses the perovskite nanocrystals seem to start to be aggregated (Appendix A). From 14.4 × 10^6^ pulses and longer irradiation duration, an anion exchange was started and thus two peaks were observed in the PL spectra (Appendix A). Similar behavior was observed for the same irradiation time, irradiating the perovskite nanocrystals without the GO (Appendix A). The initial peak remains at the same wavelength and decreases with the number of pulses while a new peak was emerged upon photo-triggered process in the absence of any reacting anion source [42]. The mechanism of anion exchange in Cl-containing solvent triggered by a laser has been proposed previously by Son’s group [24]. Based on this mechanism, the anion exchange was realized via electron transfer from the perovskite nanocrystals to the solvent producing halide ions. Anion exchange of the 0D (Cs_4_PbBr_6_) and 3D (CsPbBr_3_) phases, occurs at large number of pulses and could be confirmed from the careful analysis of the selected area diffraction patterns of the as-prepared and the irradiated samples (Appendix A). Acquiring the diffraction patterns from ensembles of mixed 3D/0D nanocrystals for the as-prepared nanocrystals (Appendix A) and the 14.4 × 10^6^ pulses irradiated samples (Appendix A), one can spot differences between the two patterns. The *d* spacings of the irradiated samples were decreased compared to the initial ones. They approach, but not reach the values for the chlorine 0D phase, indicating the partial anion exchange. In these irradiation conditions some of the nanohexagons remain on the GO, but the morphology of them start to modify and to become irregular (Appendix A). The perovskite nanocrystals start to aggregate, indicating desorption of the ligands from the particle surfaces (Appendix A). At 57.6 × 10^6^ pulses, nanocrystals of different shapes were observed (Appendix A). Some of the nanohexagons remain as previously but also quite spherical nanocrystals appeared from their partial fragmentation. The nanocrystals transform to nanoribbons through a laser-induced melting and subsequent recrystallization due to the laser energy absorption [43]. Very recently, similar structures have been observed through temperature-driven transformation and the rearrangement of atoms at the connecting facets [44].

## 4. Conclusions

In summary, we have reported a facile laser-induced process to fabricate perovskite-2D nanoconjugates. In particular, pre-prepared nanocrystals were mixed with the GO materials in solution and irradiated with a femtosecond laser at ambient conditions. It was observed that the density of the nanocrystals on the flakes can be controlled by the number of the irradiation pulses. Τhe laser pulses were efficient to promote the creation of functional groups on GO which allow perovskite crystals to bind to GO and cover the available flakes. Τhe irradiation can damage perovskite crystals so the dose has to be kept sufficiently low (<1000 pulses) to preserve their structure and morphology and eventually the luminescent properties. The solution of the nanoconjugates was PL-active following the irradiation process. The importance of this fast and easy process is that it presents a universal procedure to decorate 2D materials with metal halide nanocrystals. This laser-assisted process enables the creation of many conjugates with perovskite nanocrystals of various composition and shape. The nanoconjugates combine the exciting properties of the perovskite nanocrystals together with the interesting properties of the 2D materials. In addition, new physics and synergetic effects are emerging from the coupling between the two different materials. Among our future plans is to use this laser-assisted method to enhance the PL of the nanocomposites.

## Figures and Tables

**Figure 1 nanomaterials-10-00747-f001:**
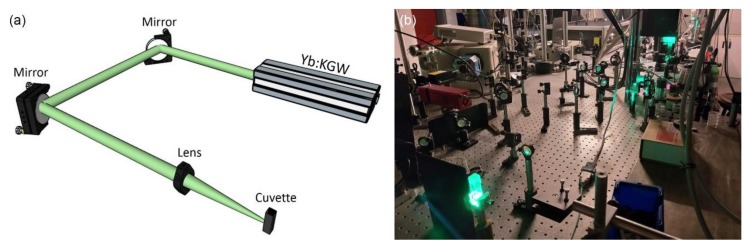
(**a**) Schematic representation and (**b**) photograph of the setup of the photo-induced process using a femtosecond laser.

**Figure 2 nanomaterials-10-00747-f002:**
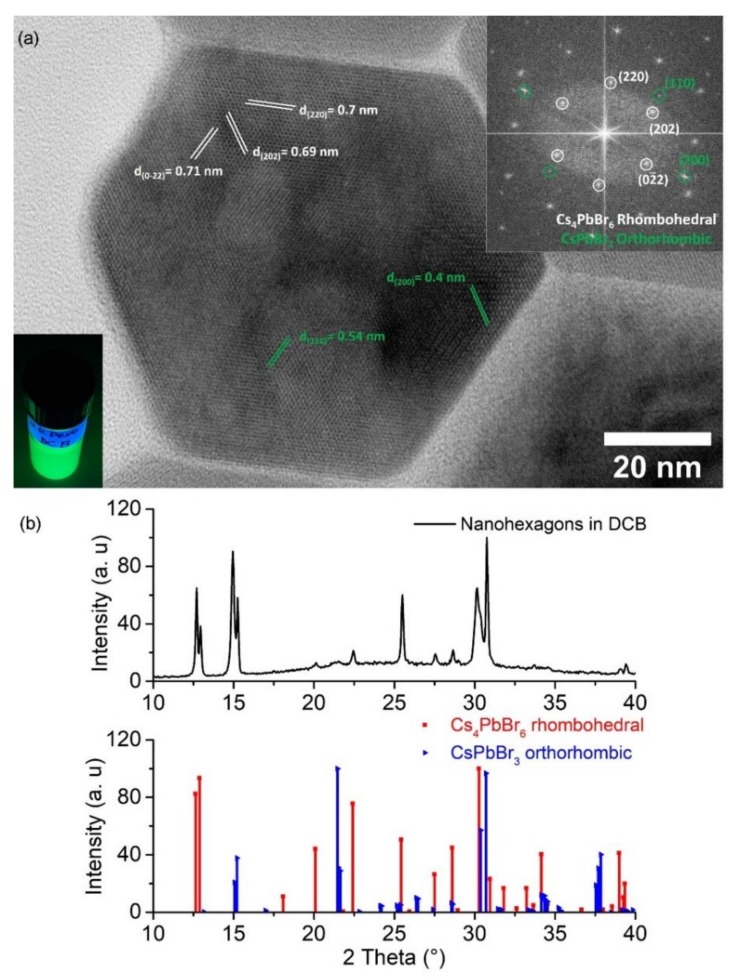
(**a**) High resolution TEM (HRTEM) image of the all-inorganic metal halide nanohexagons dispersed in dichlorobenzene. Insets: The fast Fourier transform (FFT) pattern of the HRTEM image of the individual nanohexagon (upper image) and colloidal solution photograph (under UV illumination, λ = 365 nm) (bottom image); (**b**) XRD pattern of the same sample. The reference patterns of the rhombohedral Cs_4_PbBr_6_ crystal structure (ICSD-025124) and the orthorhombic (ICSD-97851) are also provided for comparison (red and blue pattern in b).

**Figure 3 nanomaterials-10-00747-f003:**
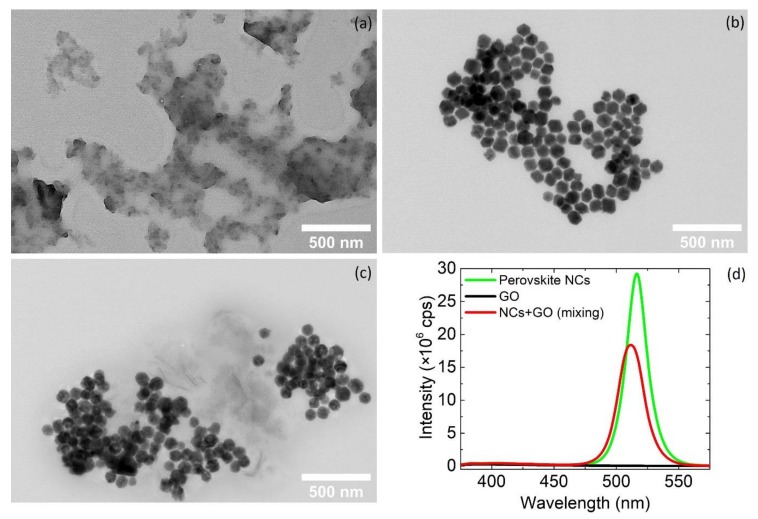
(**a**–**c**) TEM images of the exfoliated graphene oxide (GO) in 1,2–dichlorobenzene (DCB), of the metal halide nanocrystals in the same solvent and the mixture of the two before the laser irradiation; (**d**) Photoluminescence spectra of the same solutions.

**Figure 4 nanomaterials-10-00747-f004:**
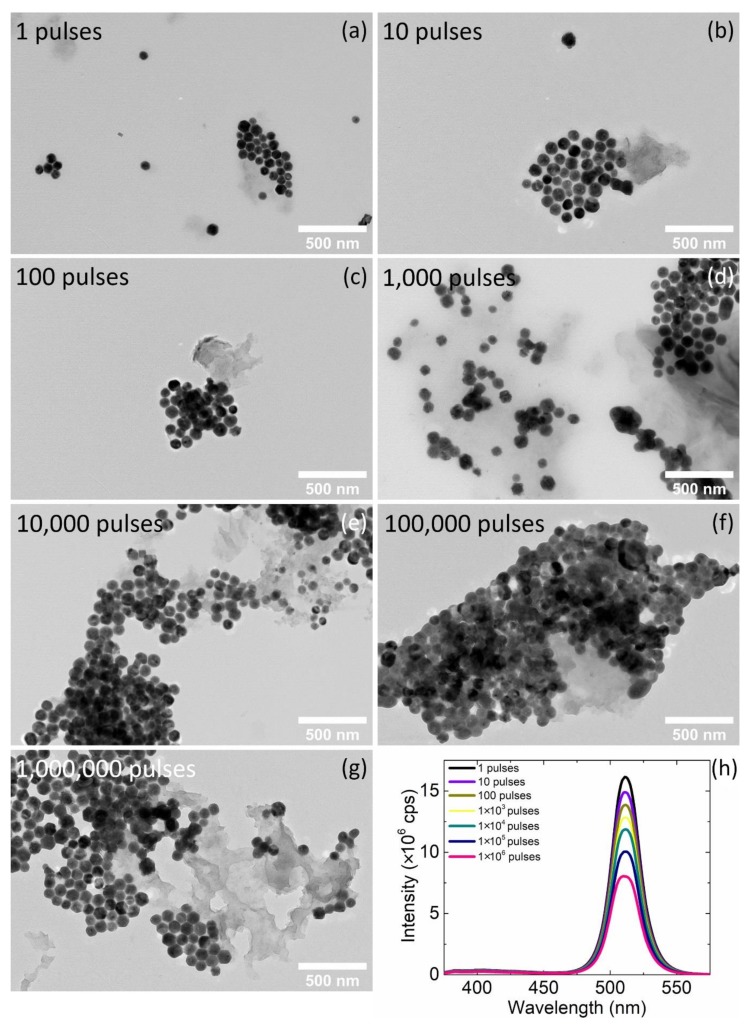
(**a–g**) TEM images of the perovskite–GO nanoconjugates after irradiation from 1 to 10^6^ femtosecond pulses; (**h**) Photoluminescence of the DCB-based irradiated solutions.

**Figure 5 nanomaterials-10-00747-f005:**
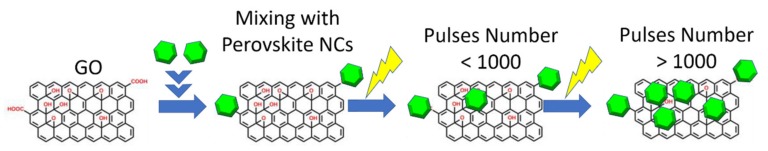
Proposed mechanism of the anchoring of metal nanocrystals on the GO flakes.

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
