# Peer review of "Laser-Assisted Fabrication for Metal Halide Perovskite-2D Nanoconjugates: Control on the Nanocrystal Density and Morphology"

_nanomaterials, 2020, doi:10.3390/nano10040747_

Round 1
Reviewer 1 Report
The paper deal with the laser assisted synthesis of GO-perovskite hybrid. The manuscript may be published after several corrections:
- Authors wrote that “The graphene oxide (GO) is also exfoliated in the same solvent (Figure 2a)” What was the source of the graphene that was exfoliated under laser irradiation?
- The XRD of the sample after 115.2 M pulses should be measured for the sample. Thus may explain the origin of the splitting of the emission band. Maybe it comes from the second phase of the perovskite. It would be also beneficial to prepare structure evolution during whole laser treatment time.
- What is the origin of the sharp dip observed at 330 nm in the absorption spectra of the composite (Fig.S3)?
- Are there any different in size of the nanocrystals after low and high number of pulses?
After explaining the above points, the manuscript can be published in Nanomaterials.
Reviewer 2 Report
The authors report on a pulsed laser-based method to decorate 2D materials with perovskite nanocrystals from a colloidal solution. Overall,I found the approach interesting and meaningful for the community working on hybrid photonic devices, however, the structure of the work is a little confusing and there is a lack of accuracy which could be easily improved by reformulating the text with the help of a native English speaker. As an example, the conclusions begins with:" a process to fabricate hybrid perovskite-2D materials…The work rather deals on decorating 2D materials with inorganic perovskite nanocrystals!
Here are my suggestions to improve the work:
- I would add a figure illustrating the design of experiment
- About the presence of 3D and 0D, I have a certain difficulty to understand where this comes from, as far as I can see there are just two crystallographic phases in the same nanocrystal. I am not sure the term 0D is appropriate here.
- In Figure 3 the evolution of the coverage with the pulse dose is illustrated; I suggest to show a wider range of interval of pulses, figure a-d are just an obvious evolution. A clearer message should be given on recommendations to obtain the highest PL response.
- The interpretation about "anion exchange is achieved via electron transfer from the perovskite nanocrystals to the Cl-contained solvent producing halide ions" should be verified experimentally.
- In the conclusions it is said that the system is PL active and stable, however the stability is not discussed in the text.
- Suggested references:
https://doi.org/10.1016/j.mtener.2018.04.005
https://doi.org/10.1016/j.nanoen.2016.02.027
Regards
Round 2
Reviewer 2 Report
I have apreciated the modifications from the authors, I think a last effort can further improve the qualit and impacto of the work.
To my understanding the decoration strategy is the core result of the work.The design of experiment (doe) showing the strategy in a visual glance would certainly add clarity and emphasize the main result of the work. The experimental setup details can appear in Figure 3 but the doe is commonly found in Figure 1. My suggestion is to insert it in the introduction when describing the experiment: "This work uses a completely different approach to anchor already fabricated metal157 halide perovskite nanocrystals on graphene-based materials. A laser-assisted method has 158 been used for the first time for the fabrication of perovskite-2D nanoconjugates in colloids."
Check and reformulate these sentences:
L142: Furthermore, in such thin nanocrystals the optical properties 142 can be tuned by control the thickness of them (nanoplatelets).
L400 Both solutions have a strong and narrow strong absorption peak
L403 been observed in systems where these phases coexistence
L457 no complete reduction of the as-prepared GO is happened (occurs) in these irradiation conditions
L460" The carboxyl groups on the edges are anticipated to coordinate mainly with Pb+2 ions"
I think this is a key information, I do not understand the meaning of anticipated here, if it was previously shown than it would be interesting to know how (XPS?) and on which structure.
L522: "for longer irradiation duration, two peaks are observed in the PL spectra indicating that an anion exchange has been started."
The explanation of this should come first
L534 spot differences of the two patterns
(between)
In the conclusions:
L626 The crystallinity and the shape of the perovskite nanocrystals remain unaltered upon irradiation with the femtosecond laser and a small number of the pulses.
The term "Unaltered" here is inaccurate, to my understanding, the conclusion is that the laser pulses are efficient to promote the creation of functional groups on GO which allow perovskite crystals to bind to GO; to a lower extent, the radiation can damage perovskite crystals so the dose has to be kept sufficiently low to preserve their structure and eventually the luminescent properties.
Regards
